# Mobile Device Usage before and during the COVID-19 Pandemic among Rural and Urban Adults

**DOI:** 10.3390/ijerph19148231

**Published:** 2022-07-06

**Authors:** Livia Jonnatan, Cherisse L. Seaton, Kathy L. Rush, Eric P. H. Li, Khalad Hasan

**Affiliations:** 1Department of Computer Science, Mathematics, Physics and Statistics, University of British Columbia, Okanagan Campus, Kelowna, BC V1V 1V7, Canada; liviatan@student.ubc.ca; 2School of Nursing, University of British Columbia, Okanagan Campus, Kelowna, BC V1V 1V7, Canada; cherisse.seaton@ubc.ca (C.L.S.); kathy.rush@ubc.ca (K.L.R.); 3Faculty of Management, University of British Columbia, Okanagan Campus, Kelowna, BC V1V 1V7, Canada; eric.li@ubc.ca

**Keywords:** mobile device usage, COVID-19, social connection, productivity, well-being

## Abstract

Technology has played a critical role during the COVID-19 pandemic. Despite affording a safe way for people to connect with others, the potential for problematic device usage (e.g., overuse, addiction) should be considered. The goal of this study was to examine mobile device use during the COVID-19 pandemic among rural and urban people in Canada. Based on an online survey conducted in the summer of 2021 in British Columbia (*n* = 465), participants self-reported spending more hours per day (M = 8.35 h) using technology during the pandemic compared to prior (M = 6.02 h), with higher increases among urban participants (*p* < 0.001). Mobile device usage scores were highest for reasons of social connectedness and productivity, with no rural/urban differences; however, urban participants reported higher use of mobile devices for their mental well-being (*p* = 0.001), but also reported higher, continuous use (*p* < 0.001), addiction (*p* < 0.001), and detrimental impacts on their physical health (*p* < 0.001) compared to rural participants. Because urban participants were more vulnerable to mobile device overuse and addiction during the pandemic, researchers and policy makers should consider the ongoing role and positive/negative impacts of mobile device use, paying particular attention to urban populations.

## 1. Introduction

In the past three decades, mobile devices have increasingly become an integral part of everyday life. Portable devices such as smartphones, tablets, and wearable technologies (e.g., Apple Watch) offer many communication features (e.g., calls, texts), allowing people to stay in touch with friends and family members. In addition, applications (i.e., apps) are now common on mobile devices, supporting a wide range of activities such as tracking health and fitness-related information, obtaining directions, shopping, and banking. Prior research indicates that people gain significant benefits such as increased productivity at work and improved relationships with family and friends through mobile device use [1]. However, researchers have also identified various problematic usage of mobile devices (e.g., addiction, overuse) and associated psychological and health problems, such as anxiety, depression, and unhealthy sleep habits [2,3,4,5]. In a recent review, people more likely to be prone to problematic device use tended to be younger, female, and more highly educated [6].

The COVID-19 pandemic has drastically transformed people’s lives worldwide. Some have suggested that mobile devices have become even more essential during the COVID-19 pandemic to foster and strengthen social connections and overall well-being [7]. As with elsewhere, within Canada, government-mandated restrictions forced people to stay home and continue their daily activities remotely (e.g., work). These changed circumstances likely impacted mobile device use; however, little research has examined changes in Canadians’ mobile device use during COVID-19, especially differences in usage between urban and rural Canadians. Investigating the changes in mobile device use before and during the COVID-19 pandemic is critical, given prior work indicating that heavy usage of technology could negatively impact physical and mental health. Social distancing and moving to online activities are ways to reduce the virus spread; however, they can potentially introduce mobile device addiction issues, which may persist even after the pandemic ends. Knowing the impacts of the restrictions on mobile device use will give researchers, policy makers, and other experts in their fields information to better mitigate and minimize any adverse effects.

A variety of theoretical perspectives have guided empirical work related to mobile technology use and misuse [6] and studies from other countries experiencing COVID-19 lockdown restrictions have suggested that device use has increased. For example, the frequency of smartphone use increased in Japan following the first wave of the pandemic [8]. Likewise, in a small study in Spain, young adults’ mobile device use increased during lockdown (March 2020) compared to data collected prior (February 2020) accompanied by an increase in sedentary behavior and a decrease in physical activity [9]. The amount of time Flemish adults used their smartphones in Belgium increased by 28%, compared to pre-pandemic [10].

It is clear that Internet use has increased worldwide; a global survey estimated that total internet traffic increased by about 60 percent following the onset of COVID-19 [11] with access to video call applications, news sites, gaming, and home-based work contributing to this increase [12]. In Canada, the number of hours spent using the internet increased during the pandemic as 27% of adults spent 20+ hours per week online compared to 19% in 2018 [13], with urban Canadians reporting higher increases compared to their rural counterparts [14]. These results suggest that COVID-19-related regulations may have triggered excessive mobile device use. Yet, other research points to a decrease in mobile device use amid an increase in WiFi usage, as North American adults stayed home/worked from home [15].

There are well-known infrastructure challenges in rural communities that may influence mobile usage. For example, there is evidence showing differences in broadband Internet speed in rural and urban areas. Only 53.4% of rural communities in Canada currently have access to 50 Mbps broadband internet speeds (compared to 89.5% of Canada overall) [16]. Furthermore, rural/urban disparities grew during the pandemic, as urban Canadian communities’ high-speed internet doubled to 51.5 Mbps in 2020, whereas in rural communities it plateaued at 5.5 Mbps [17]. Despite broadband inequities, 97.4% of rural communities have access to Mobile LTE [16] and 86% of Canadians own a smartphone [18].

Although studies have explored mobile device usage and its impact on people’s lifestyles during the COVID-19 pandemic, little is known about the pandemic’s impact on rural and urban people’s mobile device usage and its influence on their social connection, productivity, and well-being. Although mobile devices might be the solution to mitigate the disadvantages of social distancing [7], overuse may still have negative impacts. Greater engagement with mobile devices during the COVID-19 pandemic has had negative impacts, such as device addiction, depression, anxiety [19], and inactivity [9] that can contribute to more serious health problems (e.g., cardiovascular disease) [20,21,22]. In a recent COVID-19 study in France, increased screen time was the most prevalent unhealthy behavior observed, and urban-dwelling participants were nearly twice as likely to report increased unhealthy behaviors (such as screen time, but also increased alcohol intake, smoking, lower sleep quality, etc.) compared to rural participants [23].

Therefore, the goal of this study was to explore mobile device usage among people living in rural and urban areas. A mobile device in this study was defined as a device that allows the user to perform many of the same tasks as a standard computer but with the use of a touchscreen instead of a physical keyboard and mouse. The key research questions this study aimed to address included:

(1) What are the differences between rural and urban Canadians’ hours of using technology before and during the COVID-19 pandemic? (2) What are the differences between rural and urban Canadians’ frequency of mobile device use for different activities (e.g., communication, entertainment, etc.) before and during the COVID-19 pandemic? (3) What were the advantages and disadvantages rural and urban Canadians encountered using mobile devices during the COVID-19 pandemic? (4) How did rural and urban Canadians’ mobile device proficiency relate to mobile device usage and positive/negative impacts?

Within each research question we also conducted exploratory supplemental analyses to give better context to the main results (e.g., working/studying from home) and explore potential socio-demographic differences (e.g., age, gender). We considered possible associations between socio-demographic variables and all study variables because evidence indicates differences in rural demographics and social variables compared to those in urban populations (e.g., rural more likely to be older, retired, etc.).

## 2. Materials and Methods

### 2.1. Study Design and Recruitment

In the summer of 2021, we conducted an online survey in rural and urban areas in British Columbia (BC), Canada, to explore mobile device usage behavior before and during the COVID-19 pandemic. Accordingly, we advertised our online survey on BC communities’ social media (e.g., Facebook and Instagram community groups) and local community news channels (e.g., Kijiji, Castanet) as well as other local websites. We included eligibility questions for participants: a participant must be 19 years old or above, live in British Columbia, and use a mobile device. The online survey was available from 25 June 2021, until 5 August 2021. Incentives of five CAD 100, three CAD 200, and one CAD 400 draw prizes were offered to promote participation. Ethics approval was received from the research team’s home institute (#H20-01166) and all participants provided informed consent online prior to completing the survey.

### 2.2. Questionnaire

We constructed an online questionnaire focusing on demographic information, mobile device proficiency, mobile device usage, social connectedness, productivity, mobile device addiction, continuous mobile device use, and physical and mental well-being associated with mobile device use.

*Rurality*. Participants were asked to indicate their community name and communities were categorized as being ‘rural’ or ‘urban’ based on the community classification framework presented in the BC Ministry of Health rural health services policy framework [24]. This framework classifies communities as urban versus rural/remote based on population size and level of hospital care available.

*Demographics*. We collected participants’ demographic information including age, gender, household living arrangement, type of house, city/community they live in, education, occupation for the past 12 months, and whether participants have been mostly working/studying from home during the COVID-19 pandemic, and ethnicity (10 questions).

*Hours Using Technology Before and During COVID-19*. Participants were asked to report the average number of hours per day they spent using technology (e.g., mobile devices, desktop computers, laptops) before the COVID-19 pandemic (before March 2020) and during the pandemic (March 2020–present). A change in hours spent using technology score was computed by subtracting hours spent before the pandemic from hours spent during the pandemic. In addition, an open-ended question was included where participants were asked “Please explain the primary reason for an increase/decrease in your technology usage during COVID-19”.

*Frequency of Mobile Device Use*. Questions were adapted from the Media and Technology Usage Scale [25] to assess participants’ frequency in using mobile devices for eight different activities before and during the COVID-19 pandemic. Participants were asked to respond on a scale ranging from 1 (never) to 10 (all the time) about how often they used their mobile devices before and during the pandemic for communication, social media, entertainment, internet browsing, health, photos and videos, utility/productivity (e.g., calculator, weather, reminders, calendar), and navigation (maps, wayfinding).

*Mobile Device Affinity and Usage*. We asked participants to rate statements related to mobile devices’ influence on their social connectedness, productivity, mobile device addiction, continuous mobile device use, and physical and mental well-being since March 2020 (18 questions) on a scale ranging from 1 (strongly disagree) to 5 (strongly agree). Items were derived based on a review of 64 papers focusing on psychological, health, and other behavior related to mobile device use and listed the most frequently asked questions under a set of themes. Several items were adapted from the Mobile Phone Affinity scale [26], the Mobile Phone Use Survey [27], and the Problematic Use of Mobile Phones (PUMP) Scale [28]. These scales were developed using similar approaches and have demonstrated validity and reliability [26,28]. Appendix C shows the mobile device questions used in the present study. Cronbach’s Alpha’s were: 0.60 for the 3-item connectedness scale, 0.60 for the 3-item productivity subscale, 0.79 for the 3-item mental well-being subscale, 0.65 for the 2-item continuous use subscale, 0.85 for the 4-item addiction subscale, and 0.79 for the detrimental impacts on physical health subscale.

*Mobile Device Proficiency*. A ten-item, five-point (1 = never tried; 5 = very easily) (e.g., “Using a mobile device I can use the on-screen keyboard to type”) Mobile Device Proficiency Questionnaire (MDPQ) [29] was included to assess participants’ digital literacy. The MDPQ has demonstrated internal reliability as well as convergent and divergent validity in previous research [29,30]. In the present study, Cronbach’s Alpha was high (0.87). The mean of all 10 items was computed as an overall measure of digital literacy, with higher scores representing higher literacy.

### 2.3. Data Processing and Statistical Analysis

Data were analyzed using IBM SPSS Statistics for Windows, Version 27.0, released 2020, Armonk, NY, USA. We report quantitative data by using standard statistical methods such as mean and standard deviation. Our main independent variables were pre-COVID versus during COVID and rural versus urban, though we also looked at potential differences according to age, education, working/studying from home, and gender. Our dependent variables are hours using technology, frequency of mobile device use for different activities, mobile device affinity and usage, and mobile device proficiency.

Paired samples *t*-tests were used to compare hours spent using technology prior to and during the pandemic. For hours using technology and mobile device affinity usage scales, independent samples *t*-tests were used to investigate rural/urban differences and whether participants were mostly working/studying from home. Pearson’s Chi-square tests were used to examine whether more rural or urban participants were working/studying from home. Wilcoxon signed-rank tests were conducted to compare within-subject differences in frequency of mobile device use for different activities (e.g., communication, social media, entertainment) before and during the pandemic. ANOVAs were used to examine gender, education, and occupation differences in change in hours spent using technology. Mann–Whitney U tests were used to examine differences in rural versus urban, occupation (working/going to school vs. retired/not working), and whether working/studying from home in mobile device proficiency and mobile device affinity and usage (ordinal scales). Kruskal–Wallis H tests were used to examine differences in education and gender in mobile device proficiency and mobile device affinity and usage. Pearson’s correlation was used to examine the association between age and hours spent using technology; Spearman’s Rho coefficients were reported for correlations involving ordinal responses scales (age and mobile device proficiency, age and mobile device affinity and usage scales, mobile device proficiency, and mobile affinity and usage scales). For all analyses, a *p* value of less than 0.05 was considered significant.

We used a thematic analysis on the open-ended question that asked the participants the reasons for their increase/decrease in mobile device usage during the pandemic. An inductive approach was used as we determined the categories based on the keywords present in the data.

### 2.4. Data Screening

Over the 6-week data collection period (25 June 2021–9 August 2021), 617 responses were collected, of which 152 were excluded for the following reasons: (i) participant non-consent (*n* = 2); (ii) under 19 years old and/or not BC residents (*n* = 40); (iii) response of “No” to “Do you use a mobile device?” (*n* = 20); (iv) incomplete surveys beyond the demographic questions (*n* = 36); and (v) repetitive answers for blocks of questions and/or illogical answers to the open-ended questions and incorrectly answering the survey bot/attention check question (“If you are a human reading this, please select strongly agree”) (*n* = 54), Thus, our data analyses are based on the responses from 465 participants.

## 3. Results

### 3.1. Sample Characteristics

Participants were almost equally distributed in rural (*n* = 208) and urban (*n* = 257) areas of British Columbia. Of the 465 participants, 348 were female, 110 were male, 6 were nonbinary, and 1 preferred not to answer. Their ages ranged between 19 and 85 years old (mean = 40.83, SD = 17.69). The participants were from diverse ethnic and educational backgrounds. Participants’ education included university degrees (47%), trades certificates/college diplomas (24%), completed high school (26%), or some high school or less (3%). The majority of participants were working or going to school (73%), whereas 26% were either retired or not working. In addition, about 52% of the participants answered that they had mostly been working/studying from home since the beginning of the COVID-19 pandemic (March 2020), 27% answered that they had not, and the rest were not working or left the question unanswered. See Appendix A for a more detailed breakdown of demographic data and Appendix B for a summary of exploratory socio-demographic comparisons on all variables.

### 3.2. Time Spent on Technology before and during the COVID-19

On average, participants self-reported spending a significantly higher number of hours using technology per day during the pandemic compared to pre-pandemic (see Table 1). Although in general the number of hours increased for both rural and urban participants, rural participants experienced a lower increase compared to urban participants in hours spent using technology per day.

The change in hours spent using technology did not differ based on gender, F(2, 455) = 0.47, *p* = 0.62, or educational background, F(5, 453) = 1.89, *p* = 0.10, but differed according to occupation, F(2, 456) = 15.04, *p* < 0.001. Participants who were working or going to school experienced a higher increase in the number of hours they spent using technology (mean = 2.73, SD = 2.66) compared to those who were retired or not working (mean = 1.19, SD = 2.55). Moreover, participants who had mostly been working/studying from home since March 2020 experienced a significantly higher increase in the number of hours they spent using technology (mean = 3.05, SD = 2.79) compared to those who were not working/studying from home (mean = 1.72, SD = 2.44), *t*(356) = 4.68, *p* < 0.001. Furthermore, urban participants were more likely to be working/studying from home (77.0%) compared to rural participants (50.7%), χ^2^ = 27.23, *p* < 0.001.

There was a negative correlation between age and change in hours spent using technology (*r* = −0.30 *p* < 0.001), indicating that younger participants experienced a higher increase than older participants in the number of hours they spent using technology during the COVID-19 pandemic.

Participants’ open-ended responses to the question about the reason for the change (either an increase or decrease) in their technology usage were coded into four main categories for increased usage and two categories for decreased usage. There are four main reasons for the majority of participants’ increased usage: to enable remote working or studying, to connect/communicate with family and friends, to alleviate boredom from staying at home, and to access information. The most common reason for the increase was due to working or studying from home, which forced participants to rely on technology devices for communication and information access, as one participant explained, “*working full time remotely has significantly increased my daily online usage*”. Many participants reported that they used technology during the COVID-19 more than ever to stay connected with their friends and/or family members as they were not able to visit due to travel restrictions or meet up face-to-face. One participant described technology as the “*only way of communication and connection with others during quarantine*”. Another common reason for increasing technology use was for entertainment purposes (such as playing games, streaming videos/movies, and browsing social media websites), as one participant described, “*More time spent at home—boredom browsing*”. Participants also reported that their technology usage increased to access to information to keep themselves up-to-date with news and information but especially with the COVID-19 pandemic, as one participant explained, “*to find additional information re COVID-19*”.

There are two main reasons for the minority of participants’ decreased usage (*n* = 33): to alleviate negative emotions and to substitute for new hobbies. Several participants reported that their technology usage decreased to deal with negative emotions. They described being anxious, depressed, and overwhelmed as “*there was too much information available*” and “*the constant influx of new depressing situations*”. Other participants described taking up hobbies such as “*read, sew, and craft*” while staying at home, which decreased their technology use.

### 3.3. Frequency of Using a Mobile Device before and during the COVID-19 Pandemic

Participants used mobile devices more frequently during the COVID-19 pandemic for communication (W = 7.47, *p* < 0.001), social media (W = 6.85, *p* < 0.001), entertainment (W = 8.72, *p* < 0.001), internet browsing (W = 8.02, *p* < 0.001), health (W = 7.25, *p* < 0.001), utility and productivity (W = 5.65, *p* < 0.001) compared to before the pandemic. In contrast, participants used their mobile devices for navigation (e.g., map apps) less frequently during the pandemic compared to before (W = −5.41, *p* < 0.001). There was no significant change in mobile device usage for photos and videos during the pandemic compared to before (W = −0.86, *p* = 0.382).

There were no significant differences in rural and urban participants’ patterns of frequency changes but patterns differed according to gender. Female participants had the same pattern as the total group, whereas male participants followed the same pattern but showed no significant change in mobile device usage for social media (Z = −1.853, *p* = 0.064). Participants mostly working/studying during the pandemic followed the pattern of the total group, whereas those not working/studying showed the same pattern but did not change substantially in mobile device usage for navigation (Z = −0.771, *p* = 0.441).

### 3.4. Mobile Device Affinity and Usage since March 2020

Mobile device affinity and usage subscale scores, as well as individual items (means/SDs), are shown in Appendix C. Overall, participants’ mobile device usage scores were high for social connectedness (e.g., keeping participants close to their friends and/or family members), although they had lower preference scores for communicating via mobile devices over face-to-face communication. Overall, participants agreed that mobile device usage during the pandemic had positively impacted their productivity and mental well-being; however, some continuous use and addiction items were also rated highly, suggesting overuse could be problematic. Scores on the negative impacts on physical health items were also high.

#### 3.4.1. Rural/Urban Differences

Table 2 displays rural/urban differences in mobile device affinity and usage subscales. There were no differences in rural and urban participants’ use of mobile devices for social connectedness or productivity. Urban participants experienced a significantly higher positive contribution of mobile device usage to their mental well-being compared to rural participants; however, urban participants also had significantly higher scores for the continuous use and addiction scales and reported that mobile device usage had deteriorated their physical health more than the rural participants.

#### 3.4.2. Gender

Females reported higher use of mobile devices for social connectedness (M = 3.39, SD = 0.85) compared to males (M = 3.16, SD = 0.85), with no significant differences for non-binary (M = 3.61, SD = 0.65) respondents, H(2) = 7.66, *p* = 0.022. In addition, females reported higher mobile device use for mental well-being (M = 3.33, SD = 1.10) compared to males (M = 2.98, 1.13), with no differences for non-binary (M = 3.17, SD = 0.81) respondents, H(2) = 8.24, *p* = 0.016. There were no significant gender differences in productivity, addiction, continuous use, or physical health scores.

#### 3.4.3. Occupation/Work

Those working or going to school reported higher mobile device productivity (mean = 3.34, SD = 0.79) and higher continuous use (M = 3.59, SD = 1.09) compared to retired or not working respondents (M = 3.11, SD = 0.71 and M = 2.97, SD = 1.12, respectively), U = 16,306, z = −3.23, *p* = 0.001 and U = 13,937.5, z = −5.15, *p* < 0.001, respectively; however, those working or going to school also reported higher mobile device addiction (M = 2.94, SD = 1.10) and negative impacts on physical health (M = 2.59, SD = 1.19)) compared to those retired or not working (M = 2.58, SD = 1.07 and M = 1.98, SD = 1.02), respectively), U = 16,530.5, z = −3.02, *p* = 0.003 and U = 14,088.0, z = −4.90, *p* < 0.001. There was no significant difference in mobile device use for social connectedness or mental well-being by occupation.

Participants who were mostly working/studying at home had higher mobile device addiction scores (M = 3.0, SD = 1.05) and higher continuous use (M = 3.72, SD = 1.05) compared to those who were not (M = 2.71, SD = 1.15 and M = 3.21, SD = 1.16, respectively), U = 12,603.00, z = −2.29, *p* = 0.002 and U = 10,946.00, *z* = −4.08, *p* < 0.001, respectively. Those working or studying at home had a higher use of mobile devices for mental well-being (M = 3.40, SD = 1.01), compared to those who were not (M = 3.10, SD = 1.13), U = 12,681.00, z = −2.21, *p* = 0.027; however, those working/studying at home also reported a higher negative impact of mobile device use on physical health (M = 2.72, SD = 1.18) compared to those who were not (M = 2.24, SD = 1.15), U = 11,215.00, z = −3.77, *p* < 0.001. Whether participants were mostly working/studying from home since March 2020 was unrelated to mobile device use for social connectedness or productivity.

#### 3.4.4. Age

There were negative correlations between age and all mobile device affinity and usage subscales except for productivity (see Table 3), suggesting that younger participants reported higher mobile device usage compared to older participants in being connected with others and mental well-being, but also reported higher continuous use, addiction, and detrimental physical impacts of mobile device use.

### 3.5. Mobile Device Proficiency

Participants had high levels of mobile device proficiency (mean = 4.53, SD = 0.62). Urban participants had significantly higher levels of proficiency compared to rural participants (U = 20,790, *p* < 0.001).

There were significant positive correlations between mobile device proficiency and all mobile device affinity and usage subscales except for mobile device addiction (Table 3), which suggests that participants who were more proficient were using mobile devices more for social connection, productivity, and mental well-being, but also reported higher continuous use and detrimental physical impacts.

Mobile device proficiency differed with participants’ education levels, with proficiency higher for those who had completed high school (*n* = 119, mean = 4.59, SD = 0.58) or university degrees (*n* = 217, mean = 4.59, SD = 0.57) compared to other education levels, H(3) = 12.87, *p* = 0.005. Mobile device proficiency did not differ based on participants’ gender, H(3) = 2.950, *p* = 0.399. In addition, mobile device proficiency was negatively correlated with participants’ age, meaning younger participants showed higher proficiency levels than older participants (*r*_s_ = −0.34, *p* < 0.001).

## 4. Discussion

The overall increased daily use of mobile devices among rural and urban participants aligns with the global shift towards digital approaches to comply with government mandates such as lockdown and physical distancing to reduce the transmission of COVID-19. As every area of life was affected and people were forced to perform many of their day-to-day activities online, there was a dramatic increase in internet traffic [11] and a heavy reliance on technological devices such as smartphones.

Despite the overall increase in device usage, differences emerged according to geographic location. Urban participants had higher increases in usage during COVID-19 compared to rural participants (double the hours using technology during COVID-19). The higher urban usage appears to reflect both their working or studying primarily from home and thus greater reliance on technology and their higher mobile device proficiency compared to their rural counterparts. Interestingly, the higher mobile device proficiency did not mitigate the negative impacts of mobile device use; instead, it was those who were most proficient who were most likely to continuously use and experience negative physical impacts of mobile devices. Although we could not find other research examining mobile device proficiency and problematic smartphone use, proficiency is positively related to the length and frequency of mobile device use ([29]), which is consistent with our findings.

Consistent with their higher increase in technology usage (3.02 h per day), urban participants were also more prone to the negative impacts of mobile device usage, such as mobile device addiction, continuous mobile device use, and physical discomfort associated with mobile device usage. Others have called for the need for more geographic research on problematic mobile device use [31]. Yet, we could not find any other rural/urban comparisons, suggesting the present research is among the first to examine rural/urban differences in mobile device impacts during the COVID-19 pandemic.

Further, despite previous findings suggesting that a decrease in mobile device use accompanied an increase in WiFi usage for North Americans working from home [15], we found that all participants, especially urban participants and those working from home, reported increases in mobile device use for things such as communication, internet browsing, and productivity. Although participants may have been using home WiFi networks, the increased use of mobile devices to connect is consistent with the notion that mobile device use is on the rise in Canada [32].

Of concern, prior studies have also indicated negative impacts of using mobile phones on users’ physical health. For instance, poor sleep quality and blurred vision have been reported as common issues due to excessive mobile phone use [2,33]. We also observed similar results where participants reported physical discomfort associated with using mobile devices including aches and pains (e.g., neck, shoulder, etc.), light-headedness or blurred vision, and decreased sleep quantity and/or quality. Unlike previous works where females appeared to be more prone to mobile device addiction [6], we found no gender differences in addiction; however, consistent with previous works, we found that mobile device addiction was higher among participants studying/going to school and among younger participants. Previous literature has demonstrated several negative impacts of mobile device addiction including impaired cognitive capacities (e.g., executive control and working memory), difficulties with emotion and impulse regulation, and low self-esteem [34]. Recent work has also demonstrated the negative impacts of mobile device addiction on the attention and sensory networks in the brain [35], highlighting the importance of preventing or correcting mobile device addiction. Offsetting the negative impacts of mobile device usage during the COVID-19 pandemic were the positive impacts of mobile device usage. The findings from this study point to the vital role that mobile device usage played during the pandemic in helping to mitigate the side effects of the COVID-19 restrictions. Mobile devices helped users to be more socially connected, entertained, and ultimately mentally healthier. We also observed that participants in both rural and urban areas reported that mobile device usage during the pandemic helped them to be connected with their family and friends, increased their productivity, and supported them to improve their mental well-being. These findings are in contrast with previous studies that reported that a reliance on mobile devices can cause people to feel bored, lonely, and stressed [36,37,38,39].

We also found changes in mobile device usage where people in rural and urban areas reported increased usage for communication, social media, entertainment, internet browsing, health, utility, and productivity. They used mobile devices to stay connected with their friends and family members, be productive, keep themselves up to date with information, and be entertained. On the other hand, there was a decrease in mobile device usage for maps, navigation, and wayfinding. These activities are primarily connected to travel, which was restricted to a certain degree during the pandemic, explaining the decrease we observed in the data.

For technology users in general, our study results serve as a reminder that despite all the benefits a mobile device can bring, such as connecting people, mobile device usage has some negative impacts such as mobile device addiction and physical discomfort. We observed that the negative impacts are more visible to the people who live in urban areas. Consequently, the long-term negative effects of smartphone usage should be considered and balanced with the potential benefits of their use for physical distancing. As the end of the pandemic approaches, users should consider reducing smartphone usage.

Despite the valuable contribution the findings of this study make to understanding rural–urban differences in mobile device use during the pandemic, there are some limitations. First, despite attempts to recruit more male participants, three-quarters of the sample were females, hindering the generalizability of findings to male users. Second, our survey questions mainly utilized ordinal scales that had an inexact distinction between the scales, and thus were subject to individual interpretation, which is a common limitation for similar studies. Third, the study focused on a single province in Canada (BC) that may not represent mobile device usage countrywide. A future study could provide a pan-Canadian exploration of the COVID-19 pandemic’s impacts on people’s mobile device usage behavior in urban and rural areas across Canada and offer insights into regional differences. Only participants who were mobile device users were included in the study, but a comparative study with mobile device non-users would yield valuable data on potential differences in overall health/well-being/ability to stay connected during the pandemic. In the past year, the coronavirus epidemic has evolved through several phases and our 2021 study explored only two periods—prior to and after March 2020. We recommend future studies explore different phases as different behavior changes could have occurred within these phases. Additionally, this study solely considered self-reported data but future studies should consider collecting quantitative data (e.g., automatic log of usage time data) for higher data accuracy and to compare with these self-report results. Moreover, follow-up interviews could be conducted to gain an in-depth insight into the changes in mobile device usage in rural and urban areas.

Policy makers may want to reassess their regulations on shifting many essential services online, which ultimately increases people’s reliance on technology. Before the pandemic, rural people were behind in technology infrastructure. During the pandemic, rural people fell further behind as rural internet speeds plateaued [17]. The study results can help psychologists and other healthcare workers take the best necessary action to help mitigate the side effects of social distancing and increased mobile device usage on mental well-being. Mobile application designers should implement persuasive strategies to motivate users to be more physically active and reduce their mobile device usage.

## 5. Conclusions

In this paper, we present a study exploring mobile device usage among rural and urban people before and during the COVID-19 pandemic. More specifically, we conducted an online survey investigating mobile device usage and its effects on people living in rural and urban areas in British Columbia, Canada. Our findings indicated that people in rural and urban areas had a significant increase in their device use during the COVID-19 pandemic. Urban participants reported higher mobile device continuous use, addiction, and detrimental impacts on physical health compared to rural participants. More importantly, we observed that mobile devices helped people be socially connected, more productive, and mentally healthier during the pandemic. This implies that despite the harm they may cause, mobile devices played a critical role during the COVID-19 pandemic for people in both rural and urban areas of British Columbia.

## Figures and Tables

**Table 1 ijerph-19-08231-t001:** Number of hours using technology per day.

		M (SD)	*t*	*df*	*p*-Value
Average hours using technology per day	Before	6.02 (3.27)	−18.461	459	<0.001
	After	8.35 (3.78)			
Change in the number of hours using technology	Rural	1.48 (2.30)	−6.300	458	<0.001
	Urban	3.02 (2.82)			

**Table 2 ijerph-19-08231-t002:** Mobile device affinity and usage subscale scores among rural and urban participants.

	Rural	Urban	U	*p*
	(*n* = 208)	(*n* = 257)		
	M (SD)	M (SD)		
Social connectedness	3.25 (0.84)	3.40 (0.85)	29,369.5	0.064
Productivity	3.25 (0.73)	3.30 (0.81)	27,829.0	0.440
Mental well-being	3.04 (1.12)	3.41 (1.08)	31,683.5	0.001
Continuous use	3.07 (1.12)	3.70 (1.07)	35,380.0	<0.001
Addiction	2.62 (1.10)	3.02 (1.08)	32,312.5	<0.001
Detrimental impacts on physical health	2.09 (1.04)	2.71 (1.21)	34,428.0	<0.001

Note: Mann-Whitney U tests were conducted to examine rural/urban differences in mobile device affinity and usage since March 2020. Although the tests compare mean ranks for these ordinal variables, means/SDs are reported for ease of interpretation.

**Table 3 ijerph-19-08231-t003:** Correlations between age and mobile device affinity and usage subscale scores.

	Age		Mobile Device Proficiency	
	*r*	*p*	*r*	*p*
Social connectedness	−0.15 **	0.001	0.20 ***	<0.001
Productivity	−0.05	0.283	0.21 ***	<0.001
Mental well-being	−0.26 ***	<0.001	0.16 ***	<0.001
Continuous use	−0.38 ***	<0.001	0.15 ***	0.002
Addiction	−0.29 ***	<0.001	0.07	0.147
Detrimental impacts on physical health	−0.37 ***	<0.001	0.12 *	0.022

Note: Spearman’s rank-order correlation coefficients are reported. Significant codes: ‘***’ < 0.001 ‘**’ < 0.01 ‘*’ < 0.05.

## Data Availability

Data may be made available upon reasonable request to the authors.

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
