# Peer review of "Mobile Device Usage before and during the COVID-19 Pandemic among Rural and Urban Adults"

_ijerph, 2022, doi:10.3390/ijerph19148231_

Round 1
Reviewer 1 Report
This study examines the role of mobile devices during the COVID-19 pandemic in Canada. Based on a survey conducted in British Columbia, participants spent more hours per day using technology during the pandemic compared to pre-pandemic period with greater increases among urban participants. Mobile device usage scores were highest for social connectedness and productivity, with no rural/urban differences; however, urban participants reported higher use of mobile devices for mental well-being but also higher, continuous use, addiction, and detrimental impacts on physical health, compared to rural participants.
Overall the paper is well written and explores the hot of technology adoption and its adverse effects. However, I have following comments:
1) The sample is overly represented by females. Any specific reason/explanation behind it.
2) Results regarding the adverse effects of mobile usage can be discussed in more detail.
3) What implications the results might have for technology users in general? Should the mobile usage be increased or decreased?
Author Response
Please find attached the PDF file for responses to the reviewer’s comments

Reviewer 2 Report
The study is interesting. However, more scientific soundness and concrete contributions should be more emphasized. In general, this manuscript require substantial improvements. I have following comments.
- The abstract needs to be rewritten. The current version too much emphasizes the presentation of results without providing a conceptual idea and used methods. First, please clearly present objectives/purposes of this study. Then, design/methodology/approach/used methods must be indicated. For instance, how the data were collected and analyzed must be indicated. Third, important parts of the results can be presented. Finally, implications and important values of this research should be presented.
- In the introduction, it would be more appropriate to add more discussions about the importance of investigations on mobile device usage before-during the pandemic among rural and urban populations with different demographic characteristics. It is also important to provide information about sustainable use of mobile devices so that authors can see what can be problems related to changes in mobile device activities behaviors due to the pandemic.
- Research questions presented in the introduction are not entirely related to the results. Some parts of the results emphasize differences in mobile device usages among populations with different demographic characteristics such as gender, age, education, occupation, etc. Please rewrite research questions and make sure that they exactly cover what are presented in the result section. Otherwise, it will be confusing.
- This study is more like a general survey because of insufficient theoretical discussions, lack of a concrete structure and lack of a theoretical framework. Please clearly indicate what are independent variables, dependent variables, and their possible associations (theoretical assumptions). For instance, please discuss why a gender difference could affect the change in hours spent on technology, frequency in using a mobile device, mobile device proficiency, mobile device affinity and usage.
- I think the statistic results are not fully reported in the manuscript. Only some parts were selected to present. I would suggest adding the table presenting the whole statistic results either in the text or in the appendix.
- In the discussion part, it is very important to employ relevant studies or theories to confirm the results, and please make discussions based on previous studies. Otherwise, the results will not be convincing, and we cannot see the novelty of this research.
- It will be more interesting if authors discuss whether the changes in mobile device usage are good or bad, and what could be negative impacts of those changes.
- In the discussion, please provide implications of this research. It can be practical contributions for coping with negative outcomes of changes in mobile device usages.
Author Response

(The authors gave the same response as above.)

Reviewer 3 Report
Paper shows interesting study about to explore mobile device usage among people living in rural and urban areas. Mobile device in this study was defined as a device that allows the user to perform many of the same tasks as a standard computer but with the use of touchscreen, instead of a physical keyboard and mouse. Authors constructed an online questionnaire focusing on demographic information, mobile device proficiency, mobile device usage, social connectedness, productivity, mobile device addiction, continuous mobile device use, and physical and mental well-being associated with mobile device use.
But the authors should slightly modify the article.
- Data were analyzed using SPSS Version 27. Why was this instrument chosen? Authors can use for future research, for example, SAS is way more powerful than SPSS at manipulating large databases. SAS provides a way to easily access the actual statistics that are generated and allows one to manipulate them so that you end up with consistent tables not based on copy/pasting or worse manual entry. SAS has a very powerful macro language that let’s you create your own functions for repeating custom code. Or R can be used for running variety of algorithms that some times are not available in SPSS. Visualization options available in R are 1000 times better than SPSS. A model built in R can be easily deployed using R Shiny. Deployment is extremely difficult using SPSS.
- In my opinion in further steps authors should focus on
- Generalisation of proposed method.
- It's formalization.
- Positioning among other methods and comparison.
- Perhaps a more detailed description of the results of the study in the Conclusions section will improve the perception of the article.
- In what areas is it supposed to apply the presented model? How do authors see practical applications?
Author Response

(The authors gave the same response as above.)
